Estimation of environment stability for fruit yield and capsaicin content by using two models in Capsicum chinense Jacq. (Ghost Pepper) with multi-year evaluation

Baruah Joyashree 1 2
Begum Twahira 2
http://orcid.org/0000-0001-6995-3377 Lal Mohan 2 drmohanlal80@gmail.com
1 Department of Botany, Eastern Karbi Anglong College , Assam , India
2 Agrotechnology and Rural Development Division, CSIR-North East Institute of Science and Technology (NEIST) , Assam , India
Singh Anshuman
Electronic publication date: 2024 Jul 10
Publication date: 2024
Volume: 12
Electronic Location ID: e17511
Received 2024 Feb 13; Accepted 2024 May 13
Copyright: © 2024 Baruah et al.
Copyright year: 2024
Copyright holder: Baruah et al.
License: This is an open access article distributed under the terms of the Creative Commons Attribution License, which permits unrestricted use, distribution, reproduction and adaptation in any medium and for any purpose provided that it is properly attributed. For attribution, the original author(s), title, publication source (PeerJ) and either DOI or URL of the article must be cited.
License URL: https://creativecommons.org/licenses/by/4.0/

Keywords: Ghost pepper, Pungency, Crop failure, Stability, Macro-environments

Funding: GBPNIHESD, Almora as IERP project The work was funded by GBPNIHESD, Almora as IERP project. The funders had no role in study design, data collection and analysis, decision to publish, or preparation of the manuscript.

==============================
Background

Capsicum chinense Jacq. (Ghost Pepper) is well-known for its high pungency and pleasant aroma. The recent years witnessed a significant decline in popularity of this important crop due to the use of inferior planting material and lack of elite lines. To maintain constant performance across a variety of settings, it is crucial to choose stable lines with high yield and capsaicin content, as these are the most promising traits of Ghost Pepper.

Method

In this study, 120 high-capsaicin genotypes were subjected to a 3-year (kharif 2017, 2018 and 2019) stability investigation utilizing two well-known stability methods: Eberhart-Russell (ER) and additive main effects and multiple interaction (AMMI). Three replications were used following Randomized Complete Block Design for 11 traits. The experiment soil was sandy loam with pH 4.9. Minimum and maximum temperature of 18.5 °C, 17.5 °C, 17.4 °C and 32.2 °C, 31.3 °C, 32.7 °C and rainfall of 1,781, 2,099, 1,972 mm respectively was recorded for the study period.

Result

The genotype-environment linear interaction (G×E Lin.) was highly significant for days to 50% flowering, capsaicin content, fruit length and girth, fruit yield per plant and number of fruits per plant at p < 0.005. G×E interaction for fruit yield and capsaicin content in AMMI-analysis of variance reported 67.07% and 71.51% contribution by IPCA-1 (interactive principal component axis) and 32.76% and 28.49% by IPCA-2, respectively. Eight genotypes were identified to be stable with high yield and capsaicin content. The identified stable lines can be opted for cultivation to reduce the impact of crop failure when grown in different macro-environments. Moreover, the pharmaceutical and spice sectors will also be benefitted from the lines with high capsaicin content. Further research assessing the lines’ performance across various regions of India can provide a solid foundation for the crop’s evaluation at national level.

Introduction

Capsicum chinense Jacq., also known as Ghost Pepper or Naga King Chilli or Bhut Jolokia is a kind of chilli pepper certified by the Guinness Book of World Records (2006) as the hottest chilli in the world (Baruah & Lal, 2020). This important crop is native to the North East regions of India and considered as a naturally occurring hybrid of C. chinense and C. frutescence (Bosland & Baral, 2007). The crop belongs to the genus Capsicum, which is represented by twenty-five wild and five domesticated species (International Board for Plant Genetic Resources (IBPGR), 1983). The plant has got immense potential in ethno-medicinal as well as commercial point of view (Bosland, 1996; Deorani & Sarma, 2007; Baruah et al., 2014). The fruit is mostly popular for its pleasant smell and high pungency. In addition to this, the main component ‘capsaicin’ extracted from the plant has high demand for many pharmacological applications, such as treatment of neuralgic pain and different types of arthritic problems (Meghvansi et al., 2010; Ashwini et al., 2015), reducing joint pain and swelling (Sarwa, Das & Mazumder, 2014; Roy, 2015), anticancer (Yang et al., 2010; Sarpras et al., 2018), anti-inflammatory and antioxidant properties (Liu & Nair, 2010), antimicrobial property (Amruthraj, Raj-Preetam & Lebel-Antoine, 2013) and treatment of cardiovascular disorders (Zhou et al., 2010; Peng & Li, 2010).

India holds the title of leading producer and consumer of chilies in the world with main varieties being exported such as sannam, byadagi, kashmiri, teja and mundu chilli (Indian Horticulture Database, 2011; Spice Borad of India, 2023). However, there is not much information available about the export of ghost pepper, which is a native crop of North East India. It has been reported that Ghost Pepper contains more capsaicin (3–5% more) compared to the other available chilli varieties of India (Baruah et al., 2019). The Indian varieties available till date are not suitable for capsaicin extraction as they contain less capsaicin content (<1%), the normal standard needed for commercial extraction of capsaicin (De, 2003; Baruah & Lal, 2020). This capsaicin and oleoresin have high demand in national as well as world market (Borgohain & Devi, 2007; Baruah et al., 2023). Ghost Pepper with high level of pungency can overcome the barrier associated with high cost of capsaicin extraction followed by low yield. It can also become the ideal variety for extraction of oleoresin capsicum and natural red dye, which is a good substitute for synthetic colors in the food industry (De, 2003; Baruah & Lal, 2020; Baruah et al., 2023).

Although Ghost Pepper is highly popular and has potential to be the ideal chilli variety for extraction of oleoresin capsicum, capsaicin and natural red dye, it is not preferred for commercial cultivation due to its unstable performance in diverse environments. It has been reported that climatic and geographical conditions play an important role in capsaicinoids content of Capsicum species (George & Robert, 2006; Sarwa et al., 2013). Tiwari et al. (2005) reported that there is a decrease in capsaicinoids contents (50%) of Ghost Pepper if grown in other parts of India except North East India. Therefore, it is necessary to study the stability of genotypes in order to determine their true genetic potential in varying environment conditions.

For identification of stable genotypes screening of large number of genotypes should be carried out with multi-environments (Gupta, Lal & Banerjee, 2015; Begum et al., 2023; Munda et al., 2023). Till date a handful of reports exist for stability studies using the Eberhart-Russell and AMMI models in different Capsicum species (Barchenger et al., 2018; Anilkumar et al., 2018; Raghavendra et al., 2017; Cabral et al., 2017; Noman, Bhuiyan & Islam, 2015; Sharma et al., 2014). This felt the need for studying the stability of this important crop having high capsaicin content for which it is known worldwide. Also, this is the first detailed report on stability study in Capsicum chinense Jacq. using both the models for three consecutive environments and incorporating large germplasm.

Materials and Methods

Experimental material

One hundred and twenty high capsaicin (>1.2% or 192,000 Scoville Heat Unit, w/w on dry weight) containing genotypes were used for the stability study (Table S1; Baruah et al., 2019). The genotypes were planted at CSIR-North East Institute of Science and Technology (CSIR-NEIST) experimental farm, Jorhat, Assam (26°44′N, 94°9′E, 94 msl) in Randomized Complete Block Design (RCBD) with three replications and each plot measuring about 2.5 m × 3 m at 60 cm × 60 cm plant to plant and line to line spacing. The seeds were treated with Bavistin (2 gm/kg seed) to get rid of fungus infection before nursery preparation was done.

Experimental conditions

The experiments were conducted for three consecutive years i.e., kharif 2017 (Environment 1), kharif 2018 (Environment 2) and kharif 2019 (Environment 3) to estimate the stability of fruit yield and capsaicin content. All the necessary agricultural practices were followed to raise a good crop. A fertilizer dose (NPK) of 120:80:60 kg/ha/year were applied, followed by application of weedicide 2 or 3 days before planting. The experimental site soil was sandy loam with acidic pH (4.9). During the 3-year investigation a minimum temperature of 18.5 °C, 17.5 °C, and 17.4 °C, maximum temperature of 32.2 °C, 31.3 °C and 32.7 °C, annual rainfall of 1,781, 2,099 and 1,972 mm, minimum relative and maximum relative humidity (%) of 68.5, 74.7, 68.4 and 97.9, 98.4, 98.9 was recorded respectively. It was also ensured that there was no water logging in the field during the testing period.

Morphological data recording

All the morpho-agronomic characterization was done following the criteria proposed by IPGRI, AVRDC & CATIE (1995) on Capsicum species. Data were recorded for 11 traits, viz- vegetative plant height (cm), number of main branches, leaf length (cm), leaf breadth (cm), fruit length (cm), fruit girth (cm), number of fruits per plant, fruit yield per plant (gm), capsaicin content (%), days to 50% flowering and days to maturity for three consecutive years. Pooled data of 3 years was used for the final stability study.

Extraction of capsaicin

Capsaicin estimation was performed in triplicates following spectrophotometric analysis (Thimmaiah, 1999), which was further validated using uHPLC method. A total of 2-gram dry chilli powder dissolved in ethanol extract (4 mL) was subjected to water bath for 3 h at 80 °C, followed by cooling in room temperature. The supernatant of each sample was filtered (Nylon 33 mm 0.45 µm filter) and a uHPLC Ultimate 3000 (Thermo Fisher Scientific, Waltham, MA, USA) system equipped with Betasil C18 column (particle106 size 3 µm, dimension 150 × 4.6 mm) was used for analysis. The column and sampler temperature were maintained at 60 °C, 20 °C respectively, with sample volume of 5 µL. Binary mixture of water-acetonitrile at a 50:50 ratio was used as mobile phase and the flow rate was 1.5 mL/min. The complete procedure described in Bhandari et al. (2021) was used for capsaicin estimation.

Statistical analysis

Morphological data of 3 years was subjected to statistical analysis using INDOSTAT software version 9.1 programme: the Eberhart-Russell and AMMI model. Eberhart & Russell (1966) was followed to study the linear (βi) and non-linear (σ²di) components of stability. AMMI model (Gauch, 1992) analysis to check genotype-environment interaction and stability was performed for two most economically important traits, viz- fruit yield per plant (gm) and capsaicin content (%). The linear regression model for stability analysis as given by Eberhart-Russell is represented as follows:

Yij = μi + βiIj + δij

where, Yij = Mean of ith genotype at jth environment

μi = Mean of ith genotype in all the environments

βi = Regression coefficient of the ith genotype on the environmental index which measures the response of the ith genotype in varying environments

Ij = Environmental index obtained as the deviation of the mean of all the genotypes at jth environment

δij = Deviation from regression of ith genotype at jth environment.

The formula for AMMI as suggested by Gauch is represented by-

Yij = µ + gi + ej+ dij where, Yij = yield of genotype

µ = grand mean

gi= mean deviation of genotype i from µ

ej = deviation of environment j from µ

Results

Analysis of variance (ANOVA)

A pooled analysis of variance (ANOVA) for 3 years’ data considering eleven characters was studied and presented in Table 1. ANOVA results showed significant values (p ≤ 0.005) for genotype and environment components for all the characters studied (Table 1). The genotypic and environmental variances when tested against G×E were also statistically significant for all the characters studied. At p < 0.005, the mean squares due to environments + (genotype×environment) were highly significant for characters-days to 50% flowering, capsaicin content, fruit length and girth, yield per plant, number of fruits per plant and days to maturity. The G×E (Lin.) components were also significant for characters, days to 50% flowering, capsaicin content, fruit length and girth, fruit yield per plant and number of fruits per plant, while plant height, number of main branches, leaf length and breadth and days to maturity showed non-significant variation. The variance due to pooled deviation was also highly significant (p ≤ 0.005) for eight characters, viz-plant height, days to 50% flowering, capsaicin content, fruit length and girth, yield per plant, number of fruits per plant and days to maturity.

Table 1 Pooled analysis of variance of 120 genotypes of C. chinense Jacq. for three years.

	DF	Plant height (cm)	No of main branch	Leaf length (cm)	Leaf breadth (cm)	Days to 50% flowering	Capsaicin content (%)	Fruit length (cm)	Fruit girth (cm)	Fruit yield/plant (gm)	No of fruits/plant	Days to maturity	
Rep. within Env.	6	5.83	0.14	0.66	0.45	0.70	0.03	0.09	0.10	7,549.60*	94.03	0.67	
Genotypes	119	104.27***	1.22***	3.32***	1.99***	86.14***	0.77***	2.51***	2.66***	57,645.81***	1,069.500***	63.68***	
Env. + (Gen. × Env.)	240	38.23*	0.52	1.29*	0.95*	17.60***	0.04**	0.25***	0.27***	7,809.90***	87.66***	18.53***	
Environments	2	1,802.47***	7.37***	92.22***	64.76***	1,533.36***	2.30***	22.51***	23.52***	572,835.80***	5,195.45***	1,597.60***	
Gen. × Env.	238	23.41*	0.46	0.53	0.42	4.87*	0.02	0.07	0.07	3,061.78*	44.74**	5.26	
Environments (Lin.)	1	3,604.93***	14.75***	184.45***	129.52***	3066.72***	4.59***	45.03***	47.04***	1,145,672.00***	10,390.91***	3195.21***	
Gen. × Env. (Lin.)	119	17.57	0.32	0.63	0.49	4.15*	0.02*	0.07*	0.08*	3,332.55**	37.27*	4.87	
Pooled deviation	120	29.00***	0.60*	0.42*	0.34*	5.54***	0.03***	0.06***	0.07***	2,767.75***	51.76***	5.59***	
Pooled error	714	5.13	0.19	0.12	0.09	0.69	0.01	0.03	0.03	862.62	9.72	0.55	
Total	359	60.13	0.75	1.96	1.30	40.32	0.29	1.00	1.06	24,329.32	413.12	33.49	
Note:

***, **, * indicates significant at 0.5%, 1% and 5% respectively; DF, Degree of Freedom; Rep, Replication; Env, Environment; Gen, Genotype; Lin, Linear.

Eberhart-Russell linear regression model for stability study

For stability study using a Eberhart-Russell (E-R) linear regression model, 3 years data was considered as different environments. Only two economically important characters of Ghost Pepper (capsaicin content and fruit yield per plant) were detail studied. According to the E-R model, an ideal genotype is the one which possesses a high mean value (µ), a unit regression coefficient (βi) and no significant deviation from regression (σ²di ~0). Genotype with high mean values and βi = 1 is said to have average stability, i.e., their performance does not change with change in environmental condition. βi > 1, the genotype is said to have less than average stability (i.e., sensitive to environmental fluctuation but adaptable to favorable environment) and βi < 1, the genotype is considered to have more than average stability (i.e., least affected by environmental change but are poorly adapted to environment).

For the trait capsaicin content an average mean value of 2.10 was obtained (Fig. 1, Table S2A). Fifty-two genotypes have capsaicin content more than the average value (i.e., 2.09). Among them, twenty-four genotypes (line-13, 18, 22, 26, 36, 37, 39, 43, 54, 57, 61, 63, 68, 70, 72, 75, 85, 87, 100, 102, 104, 110, 114 and 118) have βi < 1 and thus can be considered more than average stable. This includes three highest capsaicin containing genotypes-18 (µ = 4.127, βi = 0.67, S²Di = −0.01), 102 (µ = 4.19, βi = 0.83, S²Di = −0.01) and 70 (µ = 3.20, βi = 0.84, S²Di = 0.04). Twenty-six genotypes (line-5, 17, 19, 24, 40, 42, 45, 53, 58, 59, 60, 65, 66, 82, 88, 89, 90, 98, 99, 101, 106, 107, 108, 109, 111, and 120) showed βi > 1 and can be considered to have less than average stability, which includes two high capsaicin containing lines-60 and 107. Significant G-E interaction was observed for linear components for this character, which indicates variable performance of the genotypes with change in environments.

Figure 1 Eberhart-Russell model for capsaicin content showing mean and regression coefficient across three environments.

For fruit yield per plant (Fig. 2) three genotypes 102 (µ = 877.06, βi = 1.02, S²Di = −331.73), 18 (µ = 619, βi = 1.06, S²Di = 445.27), and 60 (µ = 630, βi = 1.08, S²Di = 286.13) with high fruit yield were found to have average stability. Three high fruit yield genotypes, viz- 118, 70 and 120 with βi less than 1 showed significant deviation from the regression value. Ten genotypes (line-26, 27, 28, 48, 94, 98, 99, 100 and 102) with high yield and regression coefficient (βi) near to unity can be considered to have more than average stability (Table S2A).

Figure 2 Eberhart-Russell model for fruit yield per plant showing mean and regression coefficient across three environments.

For the days to 50% flowering six genotypes, (line-21, 65, 70, 101, 102, and 107) are stable with more than average value. Plant height showed an average measure of 60.29 cm with six genotypes (line-10, 18, 48, 70, 84 and 120) showing stable performance (Table S2B). Similarly, for the number of main branches genotypes 31, 51, 63, 70, 82, 84, 86, 91, 105, 106, 110, 111 showed moderate response to environmental change with a mean value of 4.13. For leaf length, nineteen genotypes (line-4, 7. 46, 55, 68, 69, 77, 83, 85, 86, 90, 91, 95, 97, 100, 103, 108, 114 and 116) produced stability with more than average performance. For leaf breadth an average value of 4.95 was obtained and sixteen genotypes (line-46, 52, 55, 60, 66, 68, 77, 78, 83, 86, 90, 95, 97, 100, 103, and 110 showed stability criteria with more than average value. Fruit length showed a mean performance of 6.53 with eleven genotypes (line-4, 6, 12, 18, 43, 44, 52, 86, 90, 93, 104) fulfilling stability criteria (Table S2A). For fruit girth a population mean of 6.76 was obtained and twelve genotypes (line-4, 18, 43, 55, 72, 80, 86, 100, 104, 114 and 120) showed stable characteristics. An average value of 62.71 was obtained for number of fruits per plant and six genotypes (line-18, 66, 70, 98, 100 and 110) were found to be stable. Ten genotypes (line-30, 39, 45, 58, 63, 65, 102, 118, 119 and 120) with maximum number of fruits did not fulfill the stability criteria as the value of βi and S²Di showed significant deviation from the standard value. For days to maturity, a mean performance of 170.58 was obtained and five genotypes (line-13, 34, 41, 60 and 68) were considered as stable according to this model criterion. Based on fruit yield and capsaicin content, the genotypes 18, 70, 100 and 102 were considered stable for further evaluation.

Stability study by AMMI model

The AMMI analysis of variance (AMMI ANOVA) tested for two important and economic traits in three environments were presented in Table 2. The analysis showed that for the trait capsaicin content and fruit yield per pant (significant at p ≤ 0.005) the main effects of genotype (G), environment (E) and G×E accounted for 89.86%, 4.49%, 5.67% and 78.53%, 13.12%, and 8.34% G-E, respectively. The presence of G×E interaction was further explained by the interaction of genotypes into three interactive principal component axes (IPCAs). The IPCA 1 depicts 67.07%, 71.51% for capsaicin content and fruit yield per pant respectively of genotype and environment sum of squares. Similarly, the second component (IPCA 2) depicts 32.76%, 28.49% for the two characters respectively, whereas the third component showed no interaction. IPCA I component was found to be highly significant for the characters studied, whereas IPCA II and III were found to be non-significant. The residual sum of squares (SS) and mean sum of squares (MSS) were also found to be non-significant for all the characters studied.

Table 2 Pooled stability analysis (AMMI ANOVA) for capsaicin content and fruit yield per plant in 120 genotypes of C. chinense.

Source of variations	DF	Capsaicin content	Fruit yield per plant	
SS	MSS	Explained% of G-E SS	Explained% of G × E Interaction SS	SS	MSS	Explained% of G-E SS	Explained% of G × E Interaction SS	
Trials	359	102.34	0.29			8,734,227.00	24,329.32			
Genotypes (G)	119	91.96	0.77***	89.86	–	6,859,852.32	57,645.82***	78.53	–	
Environments (E)	2	4.59	2.30***	4.49	–	1,145,682.26	572,841.13***	13.12	–	
G × E Interaction	238	5.80	0.02***	5.67	–	728,692.42	3,061.73***	8.34	–	
IPCA I	120	3.89	0.03***	3.80	67.07	521,080.67	4,342.34***	5.96	71.51	
IPCA II	118	1.90	0.02	1.86	32.76	207,631.71	1,759.59	2.38	28.49	
IPCA III	116	0.00	0.00	0.00	0.00	0.00	0.00	0.00	0.00	
Residual	−116	−0.00	0.00	–	–	−19.97	0.17	–	–	
Pooled residual	118	1.90	0.02	–	–	207,611.74	1,759.42	–	–	
Note:

*** Significance at 0.5%.

DF, Degree of Freedom; SS, Sum of squares; MSS, Mean sum of squares.

AMMI 1 biplot for capsaicin content among 120 genotypes at three environments is presented in Fig. 3. The main effects (G, E) accounted for 94.35% of the total variance while the IPCA 1 for 3.80%, giving a total model fit of 98.15%. This indicates the model used for the study of the character has higher accuracy. Among the three environments (E1, E2, E3), E1 have positive IPCA 1 scores near to zero with average mean performance and is considered ideal environment (Fig. 3). E2 and E3 showed less than average mean performance with negative IPCA 1 scores. Twenty-five genotypes (line-1, 13, 18, 23, 32, 34, 43, 60, 61, 65, 70, 72, 84, 85, 88, 89, 90, 97, 98, 102, 104, 110, 116, 118 and 120) were found to be most stable with high capsaicin content as they have IPCA1 value near to 0. Among them, four genotypes (line-102, 18, 72, 70) contains the highest capsaicin content (Table S3A). On the other hand, thirty-nine genotypes with more than average capsaicin content were found very unstable. The AMMI 2 biplot showed nine vertex genotypes (line-17, 99, 98, 70, 33, 49, 26, 108, 88) which are the best performer for the respective environment. Of these, genotypes 98, 99, 26 and 17 contain high capsaicin but were regarded as unstable as they are far from IPCA1 origin, whereas genotype 18 and 120 were considered to be the winner for E1.

Figure 3 Biplot (AMMI 1 and 2) for capsaicin content across three environments.

Similarly, for fruit yield per plant a total model fit of 97.61% was obtained for this character, indicating high accuracy of the studied model. E1 was found to be an ideal environment as it had IPCA 1 score near to zero with more than average performance (Fig. 4). Twenty-seven genotypes (line-5, 18, 19, 28, 34, 39, 54, 58, 60, 66, 68, 70, 82, 85, 88, 90, 100, 102, 104, 105, 106, 109, 113, 114, 118, 119 and 120) were high yielders with high stability and seven genotypes (line-38, 45, 65, 79, 80, 111, 116) were high yielders with low stability (Table S3B). AMMI 2 biplot showed eight vertex genotypes (line-65, 81, 51, 24, 32, 4, 116, 45) which were high yielders in their respective environments. Of these genotypes 65 and 32 were far away from IPCA1 origin and therefore cannot be considered stable. For environment 1 genotype 118 and 120 reported as the winner with respect to stability and performance.

Figure 4 Biplot (AMMI 1 and 2) for fruit yield per plant across three environments.

Discussion

The selection of stable genotypes with high yield and capsaicin content is a very challenging task for Ghost Pepper breeding as its quality decreases with change in climatic conditions. For this, the performance of the genotypes and their interaction with different environment should be thoroughly studied. From breeding perspective, the most desirable genotype is the one that has broad adaptability and shows stability (Begum et al., 2023). Several studies emphasized the need of stability study for a specific location in different environments and at different years (Gupta, Lal & Banerjee, 2015; Begum et al., 2023; Munda et al., 2023). In the present study, pooled ANOVA for 3 years data considering eleven characters were taken into consideration for stability evaluation. The studied environments did not show any significance difference among them, as verified by Leven’s test of homogeneity of variance (Baruah et al., 2023). Singh et al. (2009) also emphasized the importance of total years counted for study rather than multi-location study. For selection as advance genotypes it should maintain consistent performance in all environments, location, years and for this stability study is a must. The change in genotypes performance over different environments or places is a result of genotypes interaction with environments (G×E). G×E plays a major role in identification of stable genotypes among the accessions. In the present study, highly significant mean square values (p ≤ 0.005) were obtained for environment (E) and genotype (G) main effects, which indicate the notable influence of the environment on the genotypes performance for these characters. Partitioning the sum of squares into different components revealed that mean sum of squares due to Environments + (Genotype×Environment) were significant for seven characters (p ≤ 0.005), emphasizing the presence of genotype-environment interaction (G×E) for these characters. These results agree with the results obtained by Datta & Jana (2012) for days to 50% flowering, fruit length, yield per plant and capsaicin content and Tembhurne & Rao (2013) for days to 50% flowering and yield per plant. For G×E (Lin.) components significant values were observed six characters which indicate that these characters have high variability in the germplasm taken for study and their performance across environments for these characters cannot be predicted. As per earlier reports, similar case was observed by Ozukum & Seyie (2019) for days to 50% flowering, fruit length, girth and yield per plant in Naga king chilli (Ghost Pepper). Srividhya & Ponnuswami (2010), Raghavendra et al. (2017), Barchenger et al. (2018) also observed similar variations for fruit yield per plant and Cabral et al. (2017), Anilkumar et al. (2018) for number of fruits per plant in C. annum species. The G×E (Lin.) components showed non-significant values for the rest of the characters, indicating less environment influence and their performance can be predicted. Further, highly significant values obtained in pooled deviation for the eight characters indicate the non-linear response and presence of unpredictable components of G×E interaction. Analysis of variance can only provide measure of G×E but their overall response and stability cannot be truly predicted (Alwala et al., 2010). This led to the emergence of studying genotypes responses using two parametric methods: the Eberhart-Russell and AMMI models.

The Eberhart-Russell model studied for capsaicin content showed twenty-four genotypes to have more than average stability, but they are poorly adapted to any change in the environment. Of them, genotype 18, 102 and 70 possess highest capsaicin content. Twenty-six genotypes have less than average stability as they showed significant deviation for βi values and therefore cannot be considered for stability programme. It indicates that these genotypes performance were influenced by change in environment; however, they can be adapted to favorable environment. For this character, significant G×E interaction was observed for linear components which comply with the results of Zewdi & Bosland (2000) and Datta & Jana (2012) in C. annum. The non-linear component of G×E interaction was non-significant in the present study, which is similar to the findings of Raghavendra et al. (2017) in C. annum hybrids. With regards to yield per plant, three genotypes (102, 18 and 60) showed stabilities because they were unaffected by change in environment. They also have high yield capacity and thus can be considered for selection as an elite line. Ten genotypes with more than average yield showed more than average stability criteria and thus remain unaffected by any change in performance. However, these genotypes when brought to change in environmental condition will perform less and have poor adaptability. Three genotypes with high yield showed significant deviation for βi and therefore cannot be taken for stability study. For fruit yield, both linear and non-linear components of G×E interaction were found to be significant, which indicates differential response of the genotypes in different environments. This is in conformity with the findings of Lohithaswa, Manjunath & Kulkarni (2001), Tembhurne & Rao (2013) for linear components and Wani et al. (2003) and Srividhya & Ponnuswami (2010) for non-linear components in C. annum. From the study, both linear and non-linear components of G×E interaction were found to play important roles in the expression of the studied characters. This also indicates that the three environments resulted in the differential response of the genotypes accordingly.

Further AMMI model is also very efficient for studying G×E interaction and selection of stable genotypes. This model is a combination of ANOVA for studying the main effects of the genotypes and the environment including the principal component analysis (PCA) of G×E interaction and thus very useful in determining the which-won-where pattern of the genotypes (Gauch, 1992). AMMI analysis of variance showed that capsaicin content present in the fruits was influenced by the environment tested, genotypes and their interaction (G×E). The G×E was significant (p ≤ 0.005) as its presence allows further study of the characters responsible. About 89.86% of total SS was accredited to genotypic effects, 4.49% to environment effect and 5.67 % to G×E. Similarly, for yield per plant and number of fruits per plant about 78.53% and 85.81% of total SS was attributed to genotype effects, 13.12% and 7.01% to environment effect and 8.34% and 7.17% to G×E effect. Thus, the contribution of genotypic effect was found to be larger than environment effect for all the characters. Genotypic effect was higher than the environment has also been reported by Srividhya & Ponnuswami (2011) and Wondimu & Akilu (2018) in C. frutescence and C. annum respectively. For capsaicin content and number of fruits per plant, of the three components variance due to genotypes was highest followed by variance due to G×E interaction, while for fruit yield per plant variance due to genotypes was highest, followed by variance due to environment. The further partition of G×E into three interaction component analysis showed IPCA I contributed major portion for the three characters studied while IPCA III showed zero contribution. This indicates only the first two components are responsible for cent percent G×E, which is similar to the observations recorded by Srividhya & Ponnuswami (2011) for C. annum. The contribution of first two IPCA components to genotype-environment interaction was also previously reported by Crossa, Gauch & Zobel (1990), Gauch & Zobel (1996) and Purchase, Hatting & VanDeventer (2000).

According to the hypothesis proposed by Gauch & Zobel (1996), AMMI 1 biplot created for capsaicin content produced a total model fit of 98.15%. This indicates that the AMMI model used for the statistical study of the present genotypes have high accuracy and are consistent, with very negligible error. Among the test environments E1 occupied position on the right-hand side of the mean performance axis and appears to be ideal environment. Twenty-five genotypes were considered best in E1, of which four genotypes had maximum capsaicin content (line no. 102, 18, 72, 70). The environments E2 and E3 have less than average mean performance with negative IPCA 1 scores and are therefore not considered ideal environment. AMMI 2 biplot, also known as which-won-where pattern is created by joining the genotypes (vertex) farthest from the biplot origin to form a polygon view. According to Yan et al. (2000) and Yan & Tinker (2006) the genotypes at the vertex represents the highest capsaicin containing genotypes for a particular environment to which it falls. However, this does not define that these high yielding genotypes also have higher stability. Munda, Sarma & Lal (2020), Munda et al. (2023) and Lal et al. (2022) reported that those genotypes that are near to IPCA origin and form vertices are considered as the best stable lines and the genotypes included in the same environment also perform well with better stability performance. Two genotypes emerged as winners for E1 in terms of performance and stability (line no. 18 and 120). For yield per plant, E1 appeared to be the ideal environment in terms of performance and twenty-seven genotypes were found to be stable in this environment. Seven genotypes with high yield in this environment showed less stability as they were far from IPCA 1 origin. The rest of the environments E2 and E3 showed less than average mean performance with E2 showing positive IPCA 1 score and E3 with negative IPCA 1 score. The AMMI 2 biplot revealed eight vertex genotypes, which showed high yield performance in their respective environments. However, two genotypes, in spite of high performance, are not stable as they diverted from IPCA 1 origin. Two genotypes won in E1 in terms of high performance and stability (line no. 118 and 120). A total model fit of 97.61% for this character indicates the accuracy and consistency of the studied model.

Conclusions

High capsaicin content along with high productivity are the most important goal of Ghost Pepper grower, researchers and breeder. Considering the importance of the crop in context to national and international market, the present study was undertaken to determine how consistently the two economically important traits (i.e., capsaicin content and fruit yield) perform in variable environments. For this, two most widely used and accepted models of stability—Eberhart-Russell (E-R) and AMMI—were considered for the study. Based on the present study eight stable lines with higher yield and high capsaicin content were identified, viz- 18 (RRLJ-BJ-18), 60 (RRLJ-BJ-60), 70 (RRLJ-BJ-70), 100 (RRLJ-BJ-100), 102 (RRLJ-BJ-107), 104 (RRLJ-BJ-109), 118 (RRLJ-BJ-129) and 120 (RRLJ-BJ-131). These stable lines were expected to show consistent performance across a wide range of environments, provided by all the requisite agricultural practices. Further, the cultivation of these lines is aimed to reduce the impact of crop failure in different environment. To the best of our knowledge this is the first detailed stability study using both the Eberhart-Russell and AMMI models with a 3-year evaluation and incorporating large number of germplasms from its center of origin.

Supplemental Information

Supplemental Information 1 Supplementary tables.

Supplemental Information 2 Average data of three years for capsicum.

The author is thankful to CSIR-NEIST Director, Jorhat for providing all the necessary field and laboratory facilities.

Abbreviations

ER Eberhart-Russell

AMMI Additive Main Effects and Multiplicative Interaction

IBPGR International Board for Plant Genetic Resources

IPCA Interactive Principal Component Axis

IPGRI International Plant Genetic Resources Institute

ANOVA Analysis of variance

SS Residual sum of squares

MSS Mean sum of squares

AMMI ANOVA AMMI analysis of variance

G×E Lin Genotype-Environment linear interaction

Additional Information and Declarations

Competing Interests

Author Contributions

Field Study Permissions

Data Availability

Mohan Lal is an Academic Editor for PeerJ

Joyashree Baruah conceived and designed the experiments, performed the experiments, analyzed the data, prepared figures and/or tables, authored or reviewed drafts of the article, and approved the final draft.

Twahira Begum analyzed the data, prepared figures and/or tables, and approved the final draft.

Mohan Lal conceived and designed the experiments, authored or reviewed drafts of the article, and approved the final draft.

The following information was supplied relating to field study approvals (i.e., approving body and any reference numbers):

Germplasm collection were done from the wild and also from local farmers. No permissions were taken for such collections

The following information was supplied regarding data availability:

The 3 years field data are available in the Supplemental File.

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
