# Peer review of "Estimation of environment stability for fruit yield and capsaicin content by using two models in Capsicum chinense Jacq. (Ghost Pepper) with multi-year evaluation"

_PeerJ, doi:10.7717/peerj.17511_

## Round 0.1 · original submission · Major Revisions

· Academic Editor

Major Revisions

Dear Dr. Lal

Thank you for your submission to PeerJ.

A perusal of the review reports prompts me to inform you that your study entitled 'Estimation of environment stability for fruit yield and capsaicin content by using two models in Capsicum chinense Jacq. (Ghost Pepper) with multi-year evaluation' requires a suit of major and minor revisions in order to get published.

In addition to reviewers’ comments, some of my specific suggestions are as below:

1. Abstract: You need to add briefly the further studies required in the light of present findings along with potential applications that emerge from these findings.

2. Introduction section: Lines 90-92 mention about chilli production scenario in India. Would it be possible for you cite any more recent reference in place of Indian Horticulture Database, 2011.

3. Are there any plausible reasons for very high capsaicin (3-5% more) in this species that you can mention at the appropriate place (Lines 94-96).

4. Can you mention some factors responsible for the unstable performance of this species in diverse environments (Lines 103-105), taking insights from other crop species.

5. Mention briefly about irrigation, weeding and plant protection measures (Lines 129-131).

6. In Materials and Methods, give the relevant information under two separate heads: Experimental conditions and Experimental Material.

7. Similarly, instead of a single head ‘Morphological data recording and extraction of capsaicin’, you are advised to use two separate heads for morphological and capsaicin measurements.

8. The statistical analysis parts should be further elaborated, and the equation properly written.

You are therefore advised to critically consider all the comments and suggestions and bring out necessary modifications to improve the quality of the manuscript. You need to ensure some further improvements in the Discussion section. It is worth mentioning that all the changes relating to Materials and Methods including the experimental design should be meticulous and up to the mark.

It is important to highlight that your revised manuscript will be assessed again to ensure that it agrees with the reviewers' comments.

Hope to receive the revised manuscript within the stipulated time.

Reviewer 1 ·

Basic reporting

The review article entitled, "Estimation of environment stability for fruit yield and capsaicin content by using two models in Capsicum chinense Jacq. (Ghost Pepper) with multi-year evaluation (#96473) combines knowledge and application of the methodology. Readers of the journal and other researchers working on the topic internationally may find this publication useful. However, some information is missing. Given the following considerations, I believe that this work should be published, following minor changes:
1. Describe more about Figures 3 and 4 in the discussion.
2. The conclusion is very short. Revise to include the more important findings.
3. Stick to the journal's reference format for reference.
4. The references list's formatting contains several formatting problems. Here, journal titles are published in full for some and abbreviated. Please revise as per Journal style guide.
5. For specific references, there exist DOIs. Include all potential DOI numbers in the reference list, along with more recent references.
6. In addition to the previously given comments, authors are requested to proofread the entire text again to make sure that the amended version is free of any common grammatical problems.

Experimental design

Experimental designs are OK

Validity of the findings

The review article entitled, "Estimation of environment stability for fruit yield and capsaicin content by using two models in Capsicum chinense Jacq. (Ghost Pepper) with multi-year evaluation (#96473) combines knowledge and application of the methodology. Readers of the journal and other researchers working on the topic internationally may find this publication useful. However, some information is missing. Given the following considerations, I believe that this work should be published, following minor changes:

Additional comments

1. Describe more about Figures 3 and 4 in the discussion.
2. The conclusion is very short. Revise to include the more important findings.
3. Stick to the journal's reference format for reference.
4. The references list's formatting contains several formatting problems. Here, journal titles are published in full for some and abbreviated. Please revise as per Journal style guide.
5. For specific references, there exist DOIs. Include all potential DOI numbers in the reference list, along with more recent references.
6. In addition to the previously given comments, authors are requested to proofread the entire text again to make sure that the amended version is free of any common grammatical problems.

Reviewer 2 ·

Basic reporting

Thanks for sending the MS herewith my report .

Introduction: Well-presented with all the latest references and findings.
Materials Method:
1. Give a detailed information of each field experimental technique used so that a
researcher in this field may use the same in there ares
2. Did the authors look for any variations in environmental condition.
3. State the acronym for IPGRI in full in line-136
4. Pooled data of three years was considered for the study. However, the raw data
value for some characters showed incorrect values, for instance the character
“number of main branches” how can the value for such be in decimal. The
same goes for character- days to 50% flowering, number of fruits per plant and
days to maturity. This may be typographical error pls check
Results and discussion: Result of each segmented clearly presented and in details.
Discussion seemed justified in context of the findings. However, the authors are
advised to justify the discussion with more recent references.
Conclusion: The authors are requested to provide a thorough conclusion that includes
all relevant findings.

Experimental design

well presented

Validity of the findings

Fine need some new references

Additional comments

This is a novel piece of work I have checked that this is the first study using large number of germplasm taking into considerations

Reviewer 3 ·

Basic reporting

.

Experimental design

.

Validity of the findings

.

Additional comments

I have finished reviewing the MS entitled “Estimation of environment stability for fruit
yield and capsaicin content by using two models in Capsicum chinense Jacq. (Ghost
Pepper) with multi-year evaluation”. In this study authors have identified stable line of Boot jolokiya which can be used in different breeding programme and it is novel study. I recommend this manuscript can be published in this Journal. However, it needs some major modification which are mentioned below-
1. Why the authors have focused only on two characters for the crop. As it is mentioned
in line 42 “well known………pleasant aroma” I can see the crop being famous for its
pungent principle, while it does not have any relation with higher yield.
2. Line 77 “This important crop is native to the North East regions of India”. Are the
authors sure that the crop being native to NE India because as far my knowledge
chillies have their origin in South America and Brazil being the centre of origin of
C. chinense.
3. Line 127: The authors have selected stable lines by performing field experiment for
three consecutive years at the same location. Can these stable lines perform the same?
at different locations.
4. Line 271: The authors considered eleven characters for stability study using two
popular stability model. How confident are the authors that these were the only?
characters and models necessary for stability study of the crop.
5. Line 290: Line must be modified as per earlier reports
6. The authors are also asked to revised the manuscript for any grammatical errors in the
revised manuscript.
7. One or two line must be included for conclusion part
8. References must be double checked and year of the references no 6 may be checked

---

## Round 0.2 · accepted · Accept

· Academic Editor

Accept

Dear Dr. Lal

Thank you for your submission to PeerJ.

I am writing to inform you that your manuscript - Estimation of environment stability for fruit yield and capsaicin content by using two models in Capsicum chinense Jacq. (Ghost Pepper) with multi-year evaluation - has been Accepted for publication.

Congratulations!

For instance:

EDITS LINE NO: / BEFORE / AFTER / [COMMENTS]
LINE 64: / Indian / India / [.]
LINE 64: / at / at a / [.]
LINE 95: / exported are / exported such as / [.]

Reviewer 2 ·

Basic reporting

Dear Editor

All my earlier comments has successfully resolved by the authors now the MS is ready for publication

Experimental design

Well defined and correct

Validity of the findings

Yes It is very novel study can be used for other researchers

Additional comments

No

Reviewer 3 ·

Basic reporting

.

Experimental design

.

Validity of the findings

.

Additional comments

My decision is- Accept.